
# GIS analysis of effects of future Baltic Sea level rise on the island of
# Gotland, Sweden
Karin Ebert, Karin Ekstedt, Jerker Jarsjö
Department of Physical Geography and the Bolin Centre for Climate Research, Stockholm University,
S- 10691 Stockholm, Sweden.
*Correspondence to*: Karin Ebert (karin.ebert@natgeo.su.se)
## Abstract
Future sea level rise as a consequence of global warming will affect the world's coastal regions. Even
though the pace of sea level rise is not clear, the consequences will be severe and global. Commonly
the effects of future sea level rise are investigated for relatively vulnerable development countries;
however, a whole range of varying regions need to be considered in order to improve the
understanding of global consequences. In this paper we investigate consequences of future sea level
rise along the coast of the Baltic Sea island of Gotland, Sweden, with the aim to fill knowledge gaps
regarding comparatively well-suited areas in non-development countries. We study both the
quantity of loss of infrastructure, cultural and natural values for the case of a two metre sea level rise
of the Baltic Sea, and the effects of climate change on seawater intrusion in coastal aquifers, causing
the indirect effect of salt water intrusion in wells. We conduct a multi-criteria risk analysis by using
Lidar data on land elevation and GIS-vulnerability mapping, which gives formerly unimaginable
precision in the application of distance and elevation parameters. We find that in case of a 2 m sea
level rise, 3% of the land area of Gotland, corresponding to 99 km², will be inundated. The features
most strongly affected are items of touristic or nature values, including camping places, shore
meadows, sea stack areas, and endangered plants and species habitats.  In total, 231 out of 7354
wells will be directly inundated, and the number of wells in the high-risk zone for saltwater intrusion
in wells will increase considerably. Some values will be irreversibly lost due to e.g. inundation of sea
stacks and the passing of tipping points for sea water intrusion into coastal aquifers; others might
simply be moved further inland, but this requires considerable economic means and prioritization.
With nature tourism being one of the main income sources of Gotland, monitoring and planning is
required to meet the changes. Seeing Gotland in a global perspective, this island shows that holistic
multi-feature studies of future consequences of sea level rise are required, to identify overall
consequences for individual regions.
## 1 Introduction
Sea level rise is currently a fact, as stated by the IPCC (2014; references therein), being a result of e.g.
the observed slow but ongoing and irreversible collapse of the West Antarctic Ice Sheet (WAIST) and
the melt of the Greenland ice sheet (e.g. Hanna et al., 2005; Meier et al., 2007; Stroeve et al., 2007).
The pace of sea level rise is increasing; according to IPCC (2014), a global mean sea level rise of
0.63 m is *likely* to occur until the year 2100, with *virtually certain* continued sea level rise after this
point (IPCC, 2014); in a millennium scale, the near-complete loss of the Greenland Ice Sheet will with
high confidence cause mean sea level rise of 7 m (IPCC, 2014). The exact pace and amount of future
sea level rise is consequently highly uncertain (Nicholls et al., 2010). In any case, projected future sea



*Submitted to Special Issue Geomorphometry NHESS*

level rise will inundate areas along the world's coasts, where we find most of our settlements and
infrastructure (e.g. Small and Nicholls, 2003; Neuman et al., 2015).  We will need to adapt. A first
step in the process of adaptation is to investigate the consequences of sea level rise, reversible and
irreversible, on nature, humans and society, including infrastructure. We present here results for a
future sea level rise of 2 m, beyond the predictions until 2100, but still clearly below the highest
possible sea level rise.
One frequent and arguably severe effect of sea-level rise on coastal regions would be seawater
intrusion into coastal aquifer systems. For the Baltic Sea and elsewhere, most studies have primarily
been concerned with effects on climate change on river discharge (Andréasson et al., 2004; Graham,
2004; Chalov et al., 2015). Fewer have addressed groundwater resources; Sherif and Singh (1999)
present one of the first studies on the effects of climate change on seawater intrusion in two coastal
aquifers. Even relatively modest increases in average sea level will change the position of the toe of
the freshwater-saltwater interface of coastal aquifers relatively far in the inland direction. Hence,
merely due to differences in density between fresh and brackish water, the thickness and volume of
coastal freshwater reservoirs can reduce considerably. For instance, a moderate rise of one to two
decimetres in the average level of the Baltic Sea may reduce the average thickness of freshwater
reservoirs by several meters in some coastal aquifers. This may put severe constraints on
groundwater use near the coastal, although for most regions, the extent of the problem is yet
unclear.
In addition to such effects on water quality water supply, climate-driven sea level rise is expected to
have various effects on agricultural, industrial and service sectors. Flooded industrial and agricultural
land may be associated with significant production losses. Such land may potentially also release
contaminants and nutrient from the soil surface. The inherent complexity of natural and social
systems makes it a research challenge to more comprehensively understand and address the impacts
of climate change on the basis of social relevance, systemic risks and options for action. In this
context it is important to build a knowledge base that allows consideration of various conflicting
perspectives when dealing partly emotive issues (Raymond et al., 2010). Climate adaptation involves
the management of shared natural resources, where the different possible priorities and conditions
make sustainable management relatively complex. In particular, the system risks, unlike traditional
risks, need attention. Systemic risks are characterized by a high degree of complexity and uncertainty
and are usually not limited to a single sector, which requires more holistic, reflective and adaptive
strategies (Renn et al., 2011).
To allow for system analysis on the necessary overall level, we will here process spatially distributed
data and through GIS synthesize and visualize areas that are at high risk of suffering from the effects
(basic methodology e.g. according to Persson et al., 2011). Risks related to climatology,
geomorphology, hydrology, natural resources, ecology, and environmental assessment can then be
explicitly considered. We acknowledge that several studies have considered the impact of sea level
rise on a variety of environmental and anthropogenic features (Kolt et al., 2003; Blankespoor et al.,
2014) yet often they fail to take on the multi-consequential characteristic of this subject.
Furthermore, commonly the effects of future sea level rise are investigated for development
countries (Dasgupta et al., 2009; Dwarakish et al., 2009). Although such countries may be relatively
vulnerable, a whole range of regions with different characteristics need to be considered in order to
improve the understanding of global consequences. Using the case study of the Baltic Sea island of



*Submitted to Special Issue Geomorphometry NHESS*

Gotland, Sweden, we aim at filling a knowledge gap regarding comparatively well-suited areas in
non-development countries, where means may be stronger to prepare for future sea level rise, but
where at the same time historical investments in infrastructure, industries and private properties
may be considerable.
We here investigate the effects of future sea level rise on a multitude of features combined, thereby
providing a basis for assessments on overall impacts on the environment and infrastructure of the
island. We base the investigation on GIS-vulnerability mapping as a variant of the coastal
vulnerability index (CVI) (Gornitz, 1990; Dwrakish et al., 2009), addressing the following main issues:
1. Quantitatively assess some consequences of future sea level rise of 2m around Gotland
for infrastructure, cultural and environmental values
2. Establish the effect pattern on groundwater hydrology and wells
3. Establish the risk of well salinization with the current (2015) sea level and with a 2 m sea
level rise
With the results at hand will discuss which of these losses are irreversible and which might be
mended or prevented; possible economic consequences, arising from movement of humans,
movement of infrastructure, restoration of polluted areas, saline wells, decreasing tourism; possible
environmental consequences – pollution, saline wells and groundwater, decreasing area of beach
meadows and bird life, higher population density on remaining land surface; resulting consequences
for live quality: deteriorating water quality, smaller land surface, pollution, freshwater supply; if we
see Gotland as a "miniature world" – what effects have sea level rise globally?

## 2 Gotland – study area description

Located in the Baltic Sea about 80 km east of Sweden, Gotland is the county's largest island (Figure 1)
with an area of 3 140 km² and a permanent population of just under 60 000 people (Region Gotland,
2014). Climate here is temperate and characterized by its coastal position with a range of average
temperature from -2,5°C in February to 16 °C in July. Precipitation averages 500 to 700 mm/yr in the
coastal and inner regions respectively. This island setting, in combination with the distinguishable
Silurian limestone bedrock, creates key habitats for both flora and fauna that are unique to this
region. More than 8 % of the island is under official nature protection. Accordingly, cultural
landscapes and heritage on the island are rich with hundreds of stone ships (oldest dating 1000 B.C.)
and rune stones (oldest dating year 400), 92 medieval churches and extensive historical pasture- and
farm land (Region Gotland, 2014).
Tourism is in accordance an important factor in commercial life on the island. Almost 100 000
international tourists visited the island in 2014 (Region Gotland, 2014). Other important business
areas are lime and cement mining, agriculture and food industry; more than 10 % of the sheep and
lambs in Sweden are found on this island for example. Administratively Gotland constitutes both a
county and a municipality of its own; with a gross regional product per capita of 308 000 SEK in 2013
which corresponds to 78 % of the national average. Infrastructure on the island is well established:
the public transport network for example, covers most of the island and frequent ferry and airplane
departures connect it to the mainland, in 2013 40 % of the electricity supply came from wind power
stations on the island, 50 % of the rural population has access to fiber broadband and finally the
municipal water supply system holds good quality. There are issues of pollution, mainly from poor



individual sewage systems, in many of the separate water catchments however (Region Gotland,
125 2014).

Gotland was covered by Fennoscandian ice sheets during the Quaternary (Kleman et al., 1997) and is
still uplifted in glacio-isostatic adjustment, c. 1,5 mm/year in its northern parts and c. 0,5 mm/year in
its southern parts (Ågren and Svensson, 2011), which is insufficient to amend the consequences of
predicted sea level rise.
The topography of Gotland is shaped by the Silurian cover rock layers, dipping slightly towards the
east, and with a clear SW-NE structure forming the two highest ridges in the islands interior (Figure
1B). The lowest and thereby most vulnerable parts of the coastline are consequently along the east
coast and in between the ridges.

## 134 3 Data and methods

Analyses for this study were conducted in an ArcGIS environment using a broad range of data
sources: i) LIDAR elevation data and data on infrastructure (vector) provided by the Swedish National
Land Survey, ii) biological protective values (vector) from the County administrative board of
Gotland, iii) wells and soil types (vector) from the Swedish Geological Survey (SGU) and iv) a map of
annual average precipitation from the Swedish Meteorological and Hydrological Institute (SMHI).
Notably, the LiDAR-based 2m-resolution elevation model of Sweden gives a formerly unimaginable
precision in the application of distance and elevation parameters with a standard error in elevation
of 0.05 m and a standard error on plain of 0.25 m on average (Lantmäteriet, 2015). The Swedish
raster model is delivered in ASCII format. This 2 m ground elevation model is based on airborne laser
scanning of the terrain, with a point density of approximately 0.5-1 points per $m^2$. The ground
surface was produced through automatic classification of points in the point cloud, with known
elements (such as water polygons and buildings) used as supportive elements to remove buildings
and vegetation (Lantmäteriet, 2015).

### 148 3.1 Quantitative assessment of inundated areas

For quantifying the proportion of affected and lost assets, a successive overlay analysis was applied
combining the digital elevation model (DEM) with vector layers of infrastructure, cultural objects and
environmental values. The overlay layer of the DEM consisted of all pixels ≤ 2 m a.s.l. of the 2 m
elevation model, to model a sea level rise of a 2 m worst case scenario until 2100. The vector layers
of infrastructure included for example built-up areas, wells, roads, industrial areas and gas stations,
as well as natural and cultural heritages such as shore meadows, cultural grazing fields and rune
stones.

### 156 3.2 Well density and depth to groundwater

Well density was calculated with the point density function in ArcGIS, with the aim to visualize the
areas with accumulation of wells, and thereby areas of accumulation of settlement and
infrastructure, on Gotland. DEM elevation values were extracted for every well point. With the depth
to groundwater for each well available, we interpolated the depth to groundwater for the entire
area. We subtracted the depth to groundwater from the land surface topography to model the
pattern of the groundwater surface topography.



*Submitted to Special Issue Geomorphometry NHESS*

**3.3 Risk of saltwater intrusion in wells**
In assessing the risk of groundwater salinization of wells, we use the GIS-based RV method (Sazvar,
2007), which is a variant of multi-criteria risk analysis that have been used widely both for
groundwater protection and environmental management (e.g. Vias et al., 2005; Hossam et al., 2013)
as well as slope risk analysis (e.g. Tangestani, 2009; Pradhan, 2010; Sharma et al., 2013).
The RV-method is based on risk parameters and weighing factors that are assessed for individual
areas. The weighing factors R 1,2,3 are given with increasing importance of the parameters influence
on the salinization risk; R is multiplied with the internal risk values V of each parameter to achieve
the final risk value SRV (Eq. (1), Table 1).
$$SRV = V1 \cdot R1 + V2 \cdot R2 + V3 \cdot R3 + …. + Vn \cdot Rn = \Sigma\, Vi \cdot Ri \qquad\qquad (1)$$
where V is the Risk value and R is a weighing factor.
The resulting values are used in the GIS-overlay of the parameter layers (see input layers in figure 5,
and visual result of the overlay analysis in figure 6). In Sweden, this method has been used previously
in areas around Stockholm (Lång et al., 2006) and Gotland (Lange, 2013). The here used variant
comprises parameters that express 1) distance to coast, 2) distance to lakes, 3) soil type, 4) yearly
average precipitation, 5) elevation a.s.l. (Table 1). Gotland consists entirely of Silurian limestone; we
omit for simplicity bedrock type as a parameter, since the factor would be constant across Gotland
without effect on the outcome of the analysis. We here use values of parameters and weighing
factors that were developed for Sweden and Gotland in previous work (Gontier et al., 2003; Lång et
al., 2006; Sazvar, 2010; Lange, 2013). For Gotland's current ambient conditions, these parameters
and weighing factors (Table 1) were found to produce risk values that agree well with observed
chloride contents of existing wells (Lange, 2013).
The GIS-analysis using the R·V values (Table 1) was performed for current (year 2015) sea level and
for the area of Gotland after a 2 m sea level rise (Figure 6). For the 2m sea level rise scenario, the
area below 2 m a.s.l. was removed from the Gotland DEM. The parameters yearly average
precipitation, soil type and distance to lakes are assumed be the same also for a 2 m sea level rise.
The factors distance to coastline and land elevation were recalculated for the 2 m sea level rise
scenario. The distance to coast was recalculated for the new coastline; elevation above sea level was
recalculated emanating from the new elevations a.s.l., with 0 m a.s.l. corresponding to the current 2
m a.s.l.
**4 Results**
**4.1 Inundation of land area and infrastructure**
The main findings of the overlay analysis are presented in Table 2. The table presents values that
experience notable losses according to our calculation. An investigation of gas stations identified two
of the existing ones as inundated in the scenario. Some examples are illustrated in Figure 2.
In total, about 3 % of Gotland's current land surface, an area of 99 km², will be inundated in the
scenario of a 2 m sea level rise. In the relatively flat, southern part of the island, the expected
percentage of inundation is twice as high. Generally, between 1% and 3% of the overall infrastructure



*Submitted to Special Issue Geomorphometry NHESS*

will be inundated; roads, power lines, wells and individual buildings for example. The features that
show a higher proportional loss are understandably those that tend to be concentrated along the
coast, like wind power stations, lumber yards, camping sites and lighthouses.
As for cultural values there are mostly solitary objects affected, such as one smaller church and 11
ancient monuments. The characteristic sea stacks will suffer considerable loss (Figure 2B), which may
also be attributed with cultural values. Almost the entire sea stack area will be inundated. Walking in
between sea stacks will not be possible anymore; wave action will contribute to an increased erosion
of the sea stacks (Forsmark, 2001).
For natural reasons, the dominating proportion of the key habitats and nature protection zones are
located along the coast; these are indeed vulnerable to rising sea levels and potential losses here
reach 10 - 60 % (e.g. Figure 2C). With time these habitats may migrate landwards yet. 60% of shore
meadows (Table 2; Figure 2A) will be inundated. The Gotland shore meadows are unique habitats for
birds and endangered species (Olsson, 2008). Landward migration of these is limited by forests and
would require human intervention to persist (cf. Olsson, 2008; Cedergren, 2013).

**4.2 Well density and depth to groundwater**

Figure 3A shows that 231 wells, corresponding to about 3% of all wells, are located on ground
elevations between 0 and 2 m a.s.l. They will hence be directly affected by sea level rise through
inundation. Most of these wells, 131 out of 231, belong to summer houses and smaller farms. There
were also 30 energy wells, 8 large farm wells, 4 industry wells, 1 irrigation well and 49 wells with
unspecified usage. This reflects the overall spectrum of usage; outside cities and towns, wells are
used for summer houses and smaller farms (3837 out of a total of 7354), large farms (297 out of
7354), and irrigation and market-gardens (74 out of 7354). There are also a relatively large number of
energy wells (1068 out of 7354) and a smaller number of wells with industrial and other usages.
The well density is highest along the coastline and at elevations below 20 m a.s.l. (Figure 3B), which is
consistent with a higher population density near the coast. Except for northwest Gotland that
contains its largest town (Visby), coastal regions are flat (Figure 1B). Hence, despite the fact that
groundwater tables are frequently close to ground surface levels in the coastal regions (Figure 4A),
the absolute level of groundwater is currently near or at the sea surface level (Figure 4B). In
particular, in the areas shown in purple in Figure 4B the groundwater surface is at sea level, which
implies very high risk of salinization of wells even without sea level rise.

**4.3 Wells and risk of salinization**

Figure 5 shows individual contributions of different factors to the risk of well salinization. For each
factor, the risk is expressed as a risk value V (Eq. 1), where high values reflect elevated saltwater
intrusion risk. The risk for density-driven intrusion of relatively heavy salt water is clearly high (red
areas of Figure 5) at short distances to the coast where saline waters can readily replace fresh
groundwater when the fresh groundwater is pumped from the well.  Land surface elevation is also an
important risk factor, since elevation differences drive groundwater flows from land coast, which can
counteract density driven flows of saline water from sea to land. The fact that the annual mean
precipitation of Gotland is higher in its central parts than near the coast implies lower local recharge
of freshwater close to the coast, which contributes to higher salinization risks. Conversely, proximity



*Submitted to Special Issue Geomorphometry NHESS*

to lakes implies higher potential for freshwater replenishment, which decreases the saltwater intrusion
risk. The heterogeneous pattern of soil types across Gotland adds a patchiness in V. The salinization
risk is elevated in coarse sediments (Table 1), which in many cases are located close to the coast. In
contrast, lower-risk bare bedrock is frequently located further from the coast.
Figure 6A shows the current spatial variation in final risk of salt water intrusion (SRV, Equation 1),
resulting from all contributing factors of Figure 5. Reinforcing factors related to soil, precipitation,
topography and proximity to seawater make the current salt water intrusion risk considerably higher
in a zone that extends up to 5 kilometres from the coastline (yellow to red areas in Figure 6A).
Beyond that zone, risks exhibit a considerable spatial variability (light to dark blue shades of Figure
6A), but are generally lower albeit non-negligible. These current risks of salt water intrusion in
groundwater wells can be compared to estimated future risks given a projected sea level rise of 2m
shown in Figure 6B. As previously mentioned, the land area of Figure 6B is 3% lower than in Figure 6A
due to inundation from intruding seawater, which for instance is reflected in a wider straight to
between Gotland and the smaller island of Fårö in the northwest corner of Figure 6. Differences are
also pronounced on the southern tip of Gotland, see Figure 6C (current sea level) and Figure 6D (2m
sea level rise). In particular, the spit of land in the middle region of the insert maps is considerably
narrower in Figure 6D than in Figure 6C due to the 2m sea level rise. However, despite this shrinkage
of total land area, the areas that have high risk for saltwater intrusion are projected increase in the
future, which is most pronounced in the south, where for instance the very highest risk classes (red
to orange) extend much further in Figure 6D than for the current conditions depicted in Figure 6C.
In total, more than231 wells of Gotland will be inundated given a 2m sea level rise (Figure 7, rightmost
bar). Despite the reduced total number of wells in the future compared with today, Figure 7 shows
that the number of wells in the high risk value classes (18 to 22) will be considerably higher in the
future. The most pronounced change is predicted to occur for the highest risk value class (22), where
the number of wells will increase by 250% from 47 to a total value of 120. The second highest increase
is predicted to occur for risk value class 21, where the number of wells will increase by 150% in the
future, from 402 to a total value of 609.  Overall this shows that the percentage of high-risk wells will
increase in the future, considering the remaining part of the island.

## 5 Discussion

### 5.1 Multi-criteria risk analysis

Multi-criteria risk analysis with help of GIS-vulnerability mapping to identify risk areas of different
types has been used in numerous studies  (e.g. Vias et al., 2005;  Tangestani, 2009; Pradhan, 2010;
Hossam et al., 2013; Sharma et al., 2013). The advantages of this type of analysis is that parameters
used can be adapted to the characteristics of both the study area in question and the type of studied
risk. However, studies that rely on elevation models for mapping inundation from sea level rise have
frequently had too coarse vertical accuracy to support local decision making (Williams, 2013). The
on-going development of high-resolution Lidar-datasets including the presently used one for Gotland
contributes to removing this constraint from an increasing number of coastal regions of the world.
In the present study, we used overlay analysis to quantify direct losses by climate-driven rise in
average sea level due to land inundation, considering available layers related to infrastructural,
cultural and natural values. Recognising that increased average sea levels are associated with inland



*Submitted to Special Issue Geomorphometry NHESS*

advancement of today's freshwater- seawater transition zones, we use RV-analysis with weighted
risk values of parameters to identify future risk areas of saltwater intrusion in wells. Notably, we here
aimed at quantifying direct effects of mean sea level rise and land inundation, providing a basis for
understanding their contribution and significance relative to secondary effects of sea level rise and
other effects of hydro-climatic change on coastal regions. The latter effects include, for instance,
increased coastal erosion that might be expected as a consequence of future sea level rise, and that
might account for additional land loss (Sales, 2009), effects of changes in sea level extremes relative
to mean level (Williams, 2013), and effects of changes in patterns of precipitation and
evapotranspiration on surface water levels and groundwater recharge (Luoma and Okkonen, 2014).
Hence, in the context of overall impacts from multiple processes in coastal regions, results of
inundation-focussed studies such as ours should be seen as relatively robust estimates of minimum
effects, which may be exceeded due to the influence of parallel processes and secondary effects
(e.g., Torresan et al., 2012).

**5.2 Sea level rise – effect pattern on Gotland**


We find for the island of Gotland that, as globally (Neumann et al., 2015), the density of settlements
and infrastructure are considerably higher towards the coast. We show that this is reflected in an
increasing density of wells near the coastal stretch (Figure 3). In the case of Gotland, the most
endangered values are touristically interesting.  More than 50% of the wells in the risk zone of direct
inundation by a 2m sea level rise belong to summer houses and small farms. Additionally, more than
50% of the area of nature protection areas such as sea stacks, shore meadows and habitats for
endangered plants species that are naturally at the coast are threatened by direct inundation (Figure
2). This is a pattern known from other Baltic Sea states, for example Estonia (Kont et al., 2003).
Tourism and nature protection areas are main attractions and a major income of the economy of
Gotland so these losses will need to be addressed in future planning. This basic problem is for
instance also seen in the Caribbean islands, where main touristic attractions are located in coastal
regions that have been subject to considerable developments, but which however are vulnerable to
sea level rise and extreme weather events (Lewsey et al., 2004).
An indirect effect of inundation by a higher sea level will be the inland migration of areas with the
risk of saltwater intrusion in wells. Naturally the risk is highest along the coastline, with distance to
sea shore and elevation above sea level providing two main parameters in the overlay analysis.
However, in the case of Gotland this trend is even enforced by permeable soil types along the east
coast as well as lower amounts of precipitation that bring an even stronger increase of risk values
along the  lowland areas along the coastline of Gotland.

**5.3 Irreversible and reversible losses**


In case of a 2m sea level rise, according to the results of this study, Gotland will suffer the irreversible
loss of >50% of area unique nature values like sea stacks, bird and endangered species habitats as
well as shore meadows. Our results illustrate that in addition to direct and irreversible loss of land,
cultural values and infrastructure such as roads, industrial land, natural reserves and drinking water
wells, the remaining part of the island will be more vulnerable to salt water intrusion. The problem
could be further accentuated by pressures from increasing population and/ or population density of
the coastal zone (as the island shrinks). This means that the irreversible loss of 231 wells, according
to our study, would be followed by a near irreversible loss of 120 wells that are located in the group





*Submitted to Special Issue Geomorphometry NHESS*

of the highest risk values of our GIS-vulnerability analysis. Parallel studies have shown that sea level
rise may induce abrupt salt water intrusion events due to the existence of tipping points in coastal
aquifers (e.g., Mazi et al., 2013); for coastal aquifers of Gotland, the risk of such abrupt shifts appear
to be high since all five parameters we used in the overlay had the highest weighing values of the
highest risk parameters.
**5.4 Economic and societal consequences**
Our study gives a basis for further investigation of different indirect consequences of sea level rise,
not least economic and administrative implications. For example, present results show that a 2 m sea
level rise will result in inundation of approximately 2% of the total length of Gotland's road network.
This is on a par with projected road inundations as a result of 1 to 2 m sea level rise in coastal regions
of southern Europe (Demirel et al., 2015) and the U.S. (Koetse and Rietveld, 2009). Despite such
relatively low percentage of inundation, more detailed analyses of consequences in the U.S. case
showed that impacts on transport would be large due to network effects (Koetse and Rietveld, 2009).
Overall, many costs will appear due to necessary movement of inhabitants, movement/rebuild of
infrastructure, leakage of contaminants from inundated polluted areas, drilling of new wells that
replace saline ones, and decreasing tourism as cultural values disappear. Economic estimates need to
take into account different possible scenarios where environmental values need to be weighed
against economical means. This study can be the base for crucially needed future studies that include
local administration to take foresight action (cf. Libkovska and Zilniece, 2015).
**5.5 Environmental consequences and life quality**
For any region affected, sea level rise will not only pose economic problems but changes in
environmental conditions, with consequences of people's life quality. In the case of Gotland, saline
wells and groundwater, decreasing area of sea stacks, beach meadows, bird life and other nature
values, higher population density on remaining land surface with a deteriorating water quality will
have long-term effects. Even the minimum predicted sea level rise will consume 60% of Gotland's
protected shore meadows, a breeding place for a high variety of bird species. With the shore
meadows, sea stack areas and bird life diminishing, not only unique natural and cultural values, but
also, as a consequence, tourism will decrease.
With regard to industrial activities, the region that will be submerged by expected future sea level
rise contains contaminated industrial land and infrastructure that may have adverse environmental
effects such as gas stations. Sea level rise can hence imply that costs for mitigation measures
addressing Gotland's current environmental problems may change due to changing environmental
conditions; for completely submerged regions, costs for remediation may even become too high to
be feasible, due to an increased inaccessibility of flooded or partly flooded land. This in turn might
lead to severe environmental consequences for the already hard pressed Baltic Sea (e.g. Garmaga,
361  2012).

**5.6 A global perspective**
Sea level has risen and sunk many times and intensely during geological timescales (Haq et al., 1987);
however never before humans and settlements were present in coastal areas that will end up below
sea level. If we regard Gotland as a "miniature world" – a fully functioning society, will all



*Submitted to Special Issue Geomorphometry NHESS*

infrastructural necessities and good economic means, still we see that even this, globally seen well-
equipped area will suffer severe changes in the cause of climate-driven sea level rise. The present
study gives only a small extract of the complex scenario caused by sea-level rise on Gotland, where
e.g. the economic, environmental and life-quality consequences are not further assessed in this
effect-pattern vulnerability analysis. An increase in multi-factor research on the consequences of
climate driven sea level rise, as well as the distribution of the results to coastal municipalities, as well
as global exchange of experiences, are needed as soon as possible.

## 6 Conclusions

We performed multi-risk analysis, based on Lidar data on land elevation and GIS-vulnerability
mapping, to identify region-based consequences of future sea level rise and land inundation,
providing a basis for understanding their contribution and significance relative to secondary effects
of sea level rise and other effects of hydro-climatic change on coastal regions.
In case of a 2 m sea level rise, 3% of the land area of the Baltic Sea island of Gotland, corresponding
to ~99 km², will be inundated. The features most strongly effected, either by direct inundation or by
a decrease in size, are mostly items of touristic or nature values, including the complete inundation
of 35% of all camping places, 60% of all shore meadows, 60% of protected sea stack areas, and 53%
of endangered plants and species habitats. In addition to direct inundation of 231 out of a total of
7354 wells, the number of wells in the high-risk zone for sea water intrusion will increase
considerably, further diminishing the habitable land area of the island. Most of the effected wells are
summer houses and small farms that attract summer tourists.
With nature tourism being one of the main income sources of Gotland, monitoring and planning is
required to meet the changes. Some values will be irreversibly lost due to e.g. inundation of sea
stacks and the passing of tipping points for sea water intrusion into coastal aquifers; others might
simply be moved further inland, but this requires considerable economic means and prioritization.
Seeing Gotland in a global perspective this island shows that holistic multi-feature studies of future
consequences of sea level rise are required, in order to identify the consequences for individual
regions and to be able to take action adjusted to the particular needs of the region in question.
Studies like the present one can give the base for administrative discussions and planning.

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



*Submitted to Special Issue Geomorphometry NHESS*

Table 1. Factors included in the risk analysis for saltwater intrusion.

| Factor | Value | Risk value V | Weighted Risk value R·V |
|---|---|---|---|
| Distance to coast (m) | < 300 | 2 | 6 |
|  | 300-500 | 1 | 3 |
|  | >500 | 0 | 0 |
| Distance to lakes (m) | < 300 | 0 | 0 |
|  | 300-500 | 1 | 2 |
|  | >500 | 2 | 4 |
| Soil type | Coarse sediments | 2 | 4 |
|  | Fine sediments | 1 | 2 |
|  | Bare bedrock | 0 | 0 |
|  | NoData | NoData | NoData |
| Prec yearly average (mm) | < 600 | 2 | 2 |
|  | 600-700 | 1 | 1 |
|  | >700 | 0 | 0 |
| Elevation asl | < 5 | 2 | 6 |
|  | 5-10 | 1 | 3 |
|  | > 10 | 0 | 0 |




*Submitted to Special Issue Geomorphometry NHESS*



Table 2. Proportions of various types of infrastructure, cultural- and environmental values that would
be inundated in a 2 m sea level rise scenario. Areas are given in square kilometers (km$^2$) and lengths
in kilometers (km).

| Parameter | In current data | | | Inundated | | | Proportion, % | | |
|---|---|---|---|---|---|---|---|---|---|
| | Area | Length | Number | Area | Length | Number | Area | Length | Number |
| Gotland | 3 147.4 | | | 98.8 | | | 3 | | |
| **Infrastructure** | | | | | | | | | |
| Roads[1] | | 5 723.8 | | | 131.2 | | | 2 | |
| Individual buildings[2] | | | 16 570 | | | 520 | | | 3 |
| Power lines | | 312.3 | | | 3.1 | | | 1 | |
| Wells | | | 7 354 | | | 231 | | | 3 |
| Wind power stations[3] | | | 146 | | | 23 | | | 16 |
| Lumber yards | | | 3 | | | 2 | | | 67 |
| Camping sites | | | 20 | | | 7 | | | 35 |
| Lighthouses | | | 31 | | | 7 | | | 23 |
| **Cultural values** | | | | | | | | | |
| Farming land | 894.6 | | | 7.9 | | | 1 | | |
| Churches (smaller) | | | 10 | | | 1 | | | 10 |
| Wind mills | | | 212 | | | 3 | | | 1.5 |
| Ancient monuments | | | 1 140 | | | 11 | | | 1 |
| **Environmental values** | | | | | | | | | |
| Wetlands | 79.1 | | | 5.6 | | | 7 | | |
| Forests | 1 597.9 | | | 22.3 | | | 1.5 | | |
| Conservation areas | 311.0 | | | 59.2 | | | 19 | | |
| Shore meadows[4] | 31.0 | | | 18.6 | | | 60 | | |
| Bird nesting areas | 47.6 | | | 6.0 | | | 13 | | |
| Endangered plant species habitat (coastal) | 0.24 | | | 0.13 | | | 53 | | |

[1] of which several are mid-sections. From visual inspection total loss is about two times the inundated length.
[2] buildings (residential and others) outside "built up areas".
[3] unclear what year this data was updated exactly. There were 170 wind power stations in 2014 according to
Region Gotland (2014). All but one of the inundated stations are found along the southern coast.
[4] 68 % of the shore meadows are bordered landwards by open areas which possibly allow for habitat migration.



*Submitted to Special Issue Geomorphometry NHESS*



*Figure 1. Study area. A) Location in Fennoscandia; B) topography and major settlements; C) road*
*network and landcover.*




*Submitted to Special Issue Geomorphometry NHESS*

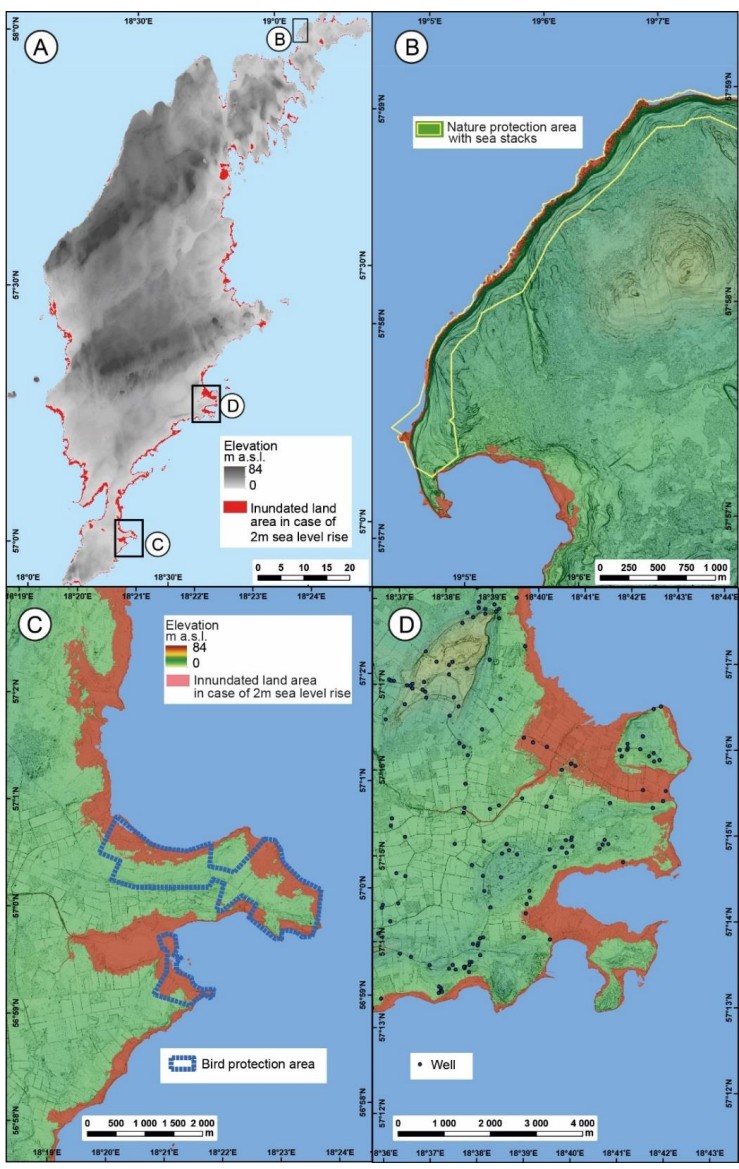


*Figure 2. A) In case of a 2m sea level rise, 99 km² of Gotland's total area of 3140 km² will be*
*inundated, corresponding to 3% of Gotland's land area. B) Gotland is famous for its nature*
*attractions, e.g. the sea stack area in Digerhuvud's nature reservat, NW Gotland. B) Gotland has the*
*richest bird life in northern Europe. A number of bird protection areas, mostly located at the coast,*
*will be inundated. C) In some areas, the sea will intrude more than 1 km inland in case of a 2 m sea*
*level rise. Except for natural and cultural values, agricultural areas and wells will be affected. In some*
*cases, new islands will be created.*

546





*Submitted to Special Issue Geomorphometry NHESS*

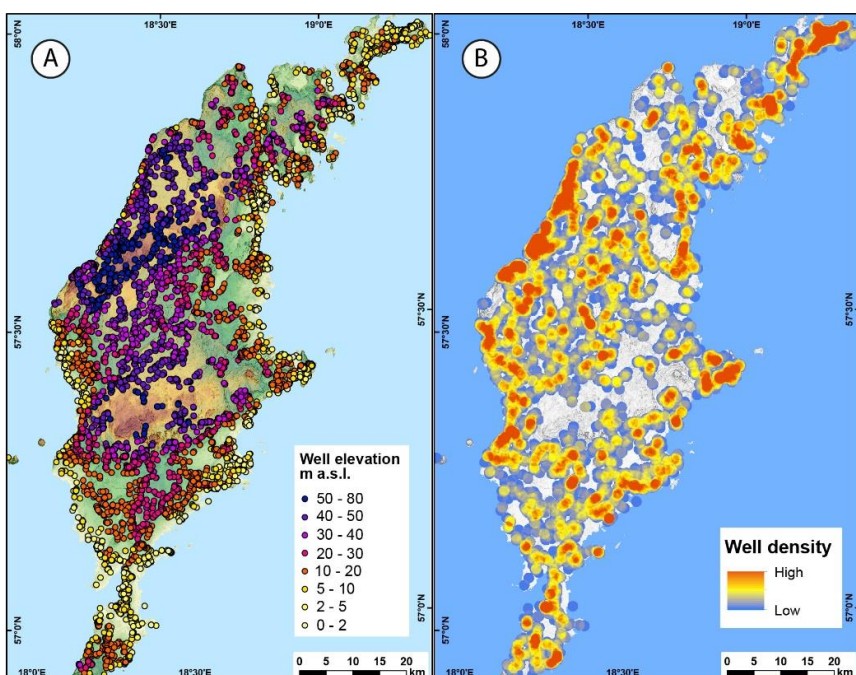

547

*Figure 3. A) 231 out of a total of 7354 wells are located on elevations 0-2 m, corresponding to about*

*3% of all wells. These wells will be directly affected by sea level rise through inundation; however*

*other wells on higher elevations further inland will be indirectly affected by salinization. B) Well*

*density expressed as point density.*





*Submitted to Special Issue Geomorphometry NHESS*


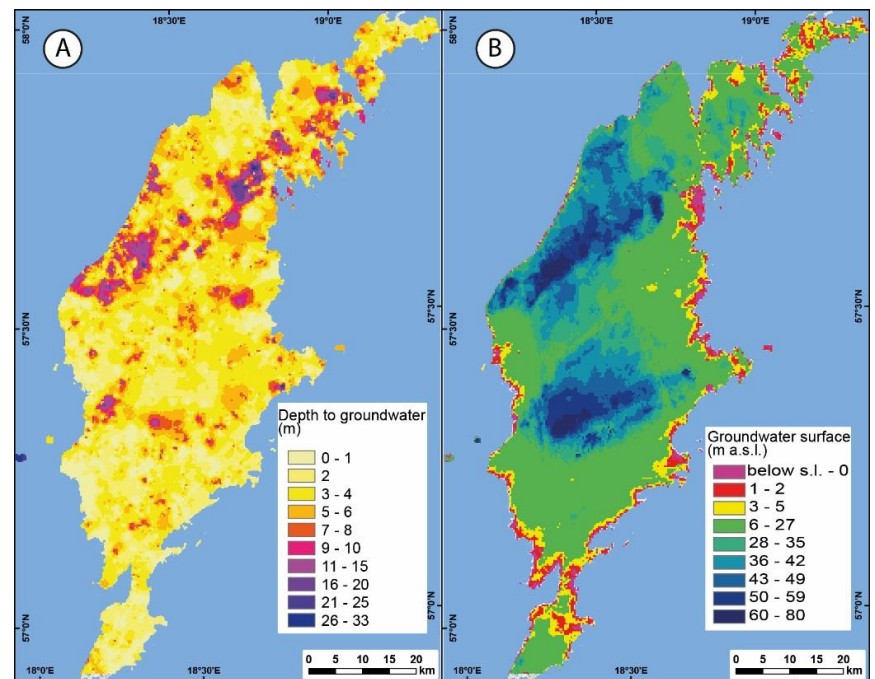


*Figure 4. A) Depth to groundwater. Values for each well were interpolated to one surface. B) The groundwater surface across Gotland, calculated by subtracting the depth to groundwater from the topography.*






*Submitted to Special Issue Geomorphometry NHESS*

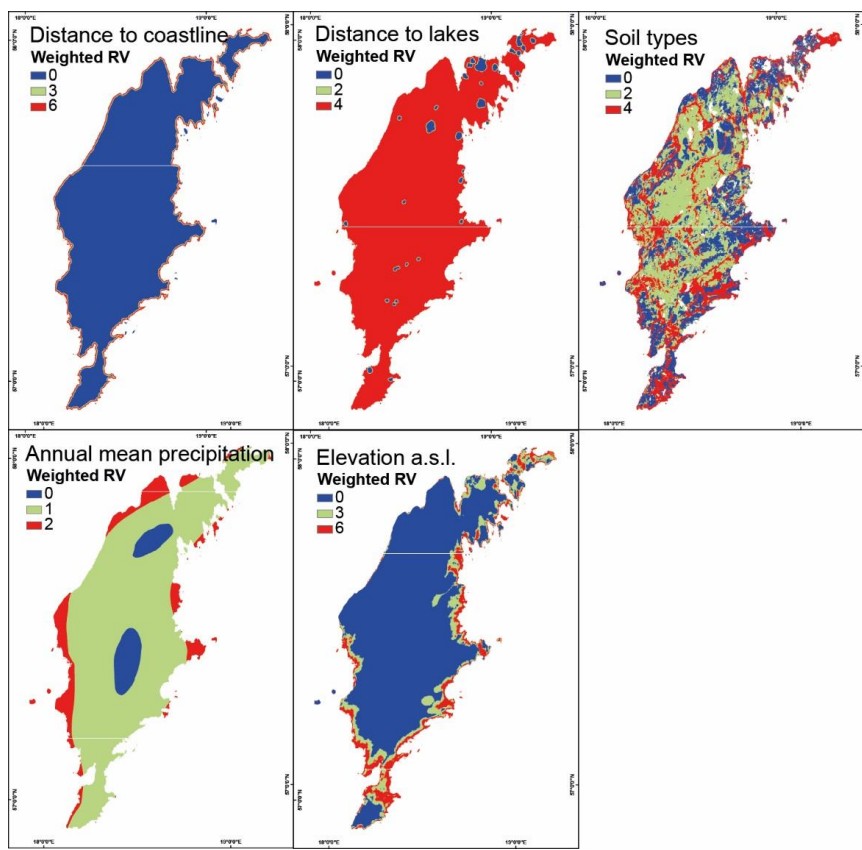


*Figure 5. Factors included in the risk analysis for saltwater intrusion (see table 2 for further*
*explanation).*



*Submitted to Special Issue Geomorphometry NHESS*

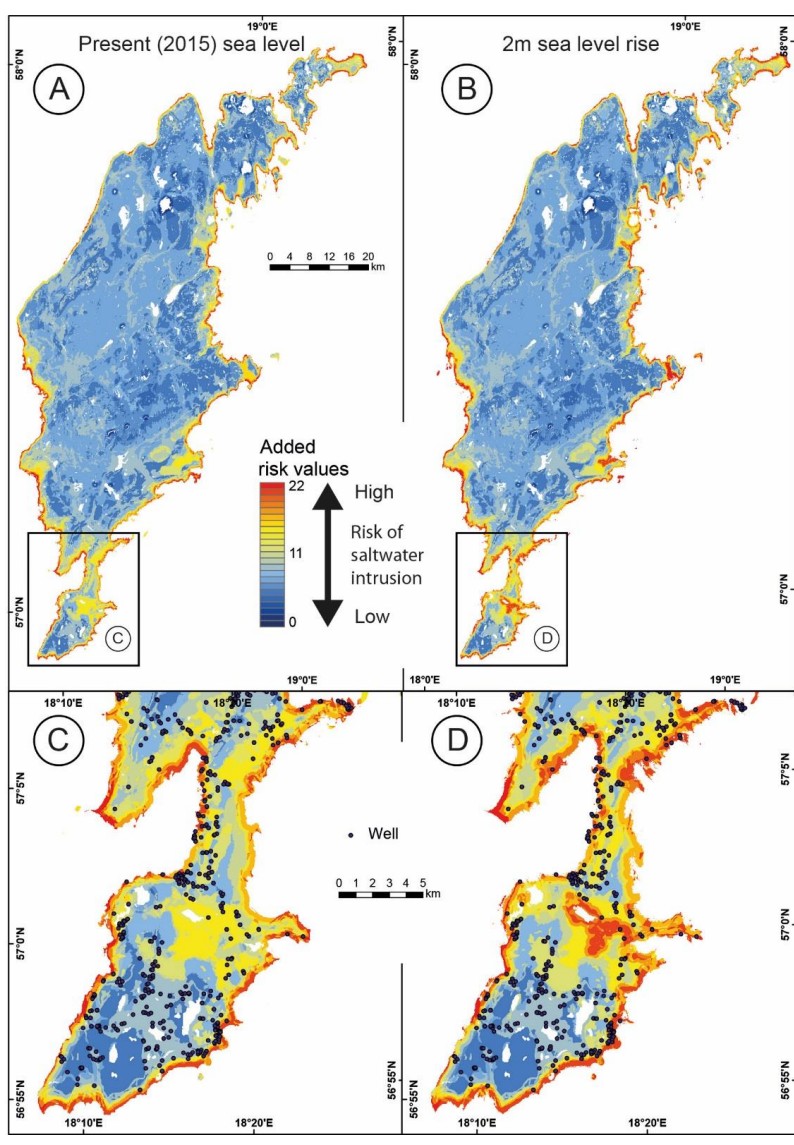


*Figure 6. Risk of saltwater intrusion as a result of the addition of weighted risk values according to*
*table 2 and figure 5. A) Risk of saltwater intrusion in wells with the current (2015) sea level; B) Risk of*
*saltwater intrusion in wells after a 2m sea level rise (areas below 2m a.s.l. are subtracted), C) and D)*
*give zoom ins of southern Gotland for both scenarios, including well locations.*





*Submitted to Special Issue Geomorphometry NHESS*

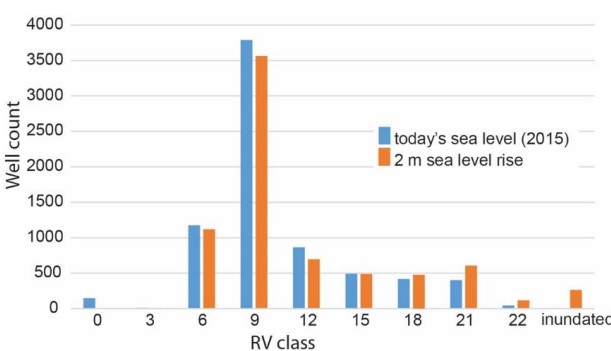


*Figure 7. Risk value classes of wells on Gotland for today's (2015) sea level and after 2m sea level rise.*

*The number of wells in higher risk classes will increase. Wells below 2 m above the current sea level*

*will be inundated.*



