# Peer review of "GIS analysis of effects of future Baltic Sea level rise on the island of"

_Natural Hazards and Earth System Sciences, 2016_

## Referee Comment (RC1) · Anonymous Referee #1 · 13 Apr 2016

General comments:

The paper deals with effects of increased sea level on the island of Gotland in the Baltic Sea. The paper is based on a GIS overlay analysis and tries to do a multi-criteria risk analysis. For future scenarios a multi-criteria risk analysis is necessary. I find the aspect of sea water intrusion to coastal aquifers due to increased sea level, and risk of saline contamination of wells as the most interesting contribution. The paper aims to fill knowledge gaps, and link the study to non-developed countries, though I find the global comparison a bit vague.

Specific comments:

Part 2. Study area description – Why Gotland? The second island of the country, Öland, is less elevated and possibly at higher risk to be flooded. I think Gotland is a

good choice but it should be elaborated. For example, the present situation on Gotland where the ground water situation is already problematic. The study should have included that the quantity of ground water is found unsatisfactory in the classification related to the EU water framework directive. Already today there is a risk of fresh water scarcity in some areas of Gotland during summer month. The lowest area, and most vulnerable parts in the study correspond to areas in the southeastern part of Gotland where the ground water quantity is unsatisfactory already according to Water Information System Sweden (http://www.viss.lansstyrelsen.se/). I think the article should have addressed this question, and the present risk of Sea water intrusion.

Part 5. Discussion – The structure of the discussion is not clear. I find it more confusing than informative with too many sub titles, some parts is also repeated (birds 5.3, 5.5). Economic values are mentioned but could be elaborated more, considering an ecosystem service view. Now, the economic and societal consequences relate to infrastructure, movement of housing, decreasing tourism, and wells. I think the life quality, bird life (5.5; 5.6) could be discussed together. The decrease of natural values ends up as economic consequences.

Technical corrections:

Line 104. County should be country. Gotland is the largest island of Sweden.

Line 365. Is it possible to say "never before", I think the wording is not correct. Settlements situated in river mouths on delta sediments are rebuilt during history. In the Hanö bay area of the Baltic Sea transgression have been studied by quaternary geologist for a long time, and there is also an ongoing archeologic project studying settlements from 11000 year ago found on the sea bed in the same area. Björn Nilsson (project leader) also published a paper with the title: Flooded Stone Age: Towards an Overview of Submerged Settlements and Landscapes on the Continental Shelf, The European Archaeologist, 2012(38), pp.84-85.

Line 531, Figure 1. A – The location map is too small, it is difficult to read the text

Sweden, Gotland etc. C – It is not possible to see the cities/towns on the map, and the electricity line is almost not visible

Line 547, Figure 3. A – It is not possible to see the difference of colors for the wells elevation 0-2; 2-5; 5-10 m.a.s.l.

---

## Referee Comment (RC2) · Anonymous Referee #2 · 26 Apr 2016

General comments: The paper deals with the impact on infrastructure, culture values and environmental values due to a sealevel rise amounting to 2 m by using accurate LIDAR level data in a GIS. The idea is good and the paper points out that the values will be heavily affected due to that much of values are located close to the coastlines. It's interesting with a multi-aspect paper but much is focused on increased salinization of groundwater. However, very little of data is presented and almost no calculations. But the work, so far is well done and this is something that the planning offices at the Gotland county should have done. It should have been nice to have a more quantitative estimation of the values or qualitative, pointing out where on Gotland the values are threatened at most. How is different values distributed and connected to the population density?

However, looking at the effects of sealevel rise it is important to consider not only land

which is inundated (as in the paper) but also land where drainage will be failed leading to less crop production. It is also inadequate just to look at median seawater level since the impact relates to the maximum height of wave actions. More extreme wave scenarios will probably be a result of the increased temperature.

The novel of the paper is the use of detailed LIDAR data for the GIS evaluation.

My concern is regarding the use of information for the RV method. It is not structurized, which it should be according to the principles behind the RV-method, instead uses factors and weights without critical discussion, handling those as if they represent a truth value of salinization risk (se more detailed discussion below). I suggest a major revision of this part and/or present data that support the use and choice of the factors.

Specific comments:

L 53-57 The statement that two dm change of the Baltic Sea-level leads to several meters loss of freshwater aquifer is wrong. Ghyben-Herzberg eq cannot be directly applied to the brackish waters in Baltic Sea due to the small density difference where other aspects such as temperature and circulation currents will be more important and lead to a very broad mixed zone. And it is not at all applicable to heterogeneous aquifers (crystalline rocks and sedimentary rocks) prevailing around the Baltic Sea. No reference for the statement is presented.

160- Experience from field measurements indicate that there is not a groundwater surface existing on Gotland but a unique groundwater pressure level at each well depending on it's specific depth, geological structures (fractures, stratigraphy, karst) etc. The level varies with season, sometimes up to 20 m. The well data gives probably information just from the time when the specific well was constructed (I can't find it mentioned in the text), since I assume it is based on the well archieve at SGU. Can we then make a groundwater surface and rely on what it represents?

The method used refers to the RV-method and the authors explain that the parameters

and the weighting used are similar to what previously been developed for Sweden and Gotland. However, the authors have unfortunately not understood the dynamics of the RV-method. The factor values should be ranked from negative to positive values since it is the extreme conditions that will have a most significant influence on the results. The RV-method also comprises an uncertainty evaluation in the same manner and factors and weighting should be based on a multistatistical evaluation of existing data from the specific type of terrains that is studied. No such statistical analysis is mentioned. How can we know that the factors selected for Gotland are the most appropriate and that the weighting is statistically correct? The method might have been inspired by the RV-method but cannot be presented as a prolongation of the method.

The papers present five factors included in the risk analysis for saltwater intrusion with reference to a non-published undergraduate report. The choice of factors is therefore unclear. However, a statistical analysis of drilled wells at Gotland made at KTH (Pirnia A, 2012, TRITA-LWR MSc degree project 12:12) shows that two of the variables used in the paper have no statistical significant to chloride wells on Gotland. Two other factors not used in the paper (type of bedrock and landuse) show instead significant correlations to chloride content in the wells. Chloride affected wells are in reality spread all over Gotland since most of the chloride comes from fossil seawater, not from recent seawater intrusion. The method used gives a too high weight to the distance from the shore. A paper submitted to Environmental Earth Sciences shows that the chloride situation on Gotland is much more widespread than is shown in Figure 6 (I enclose a statistical analyses and a GIS map based on the RV-method - not yet scientifically published -cannot be cited)

Table 1. Results of Kruskal-Wallis ANOVA by ranks for chlorid

| factors | significance level,p | classess | chloride median value mg/l |
|---|---|---|---|
| land use at well location | 0.0764 | Urban area | 16.00 |
| | | field | 49.50 |
| | | forest | 25.50 |
| soil type at well location | p=0.901 | Till | 68.000 |
| | | Sand & Gravel | 26.000 |
| | | Bedrock | 35.000 |
| | | Clay & Silt | 12.000 |
| | | Organic.material | 38.000 |
| | | | |
| Bedrock at well location | 0.0013 | Marlstone | 53.00 |
| | | Limestone | 23.00 |
| soil depth | p=0.3 | very low | 59.000 |
| | | low | 36.000 |
| | | medium | 35.500 |
| | | high | 16.000 |
| elevation | p=0.0008 | 0-20 m | 55.000 |
| | | 20-40 m | 26.000 |
| | | 40-60 m | 11.000 |
| | | >60 m | 8.300 |
| distance to saline water | 0.048 | 0-100 | 250.000 |
| | | 100-500 | 34.000 |
| | | 500-2000 | 36.000 |
| | | >2000 | 31.000 |
| slope | p=0.46 | 0-2 deg | 36.000 |
| | | 2-5 deg | 16.000 |
| | | >5 deg | |
| distance to deformation | p=0.28 | 0-50 m | 12.000 |
| | | 50-100 m | 14.000 |
| | | 100-200 m | 35.500 |
| | | 200-500 m | 56.500 |
| | | >500 m | 36.000 |

| p<0.05 | p<0.1 | p<0.2 |
|---|---|---|

**Fig. 1.**

[Figure]

[Figure]

[Figure]

[Figure]

[Figure]

**Fig. 2.**

---

## Referee Comment (RC3) · Anonymous Referee #2 · 26 Apr 2016

Continuation of RC2, specific comments:

The authors should also refer to the original presentation of the RV method in GIS, which is in Lindberg, Olofsson and Gumbrich (1996) [Lindberg, J., Olofsson, B., Gumbricht, T., (1996): Risk mapping of groundwater salinization using Geographical Information Systems. 14th Salt Water Intrusion Meeting (SWIM), 16-21 June 1996, Malmö, Sweden. SGU Rapporter och Meddelanden no 87, pp 188-197. ].

283- There are no statistical data presented that there is a transition zone that extends inland between saltwater and freshwater. Probably such zone (if exist at all) is probably only related to the coastal fringe, 0-300 m. Former investigations point out that the level of the salt groundwater is between -0 and -30 m in large areas of southern Gotland (Tullström 1954) [Tullström, H. 1954: Preliminary results from hydrogeological

investigations on Gotland. Grundförbättring no 7.(In Swedish)]

329- You can't use the weighting values as proof for the risk for abrupt changes since you don't present any data that supports the statement.

Technical comments:

The figures are generally good and informative. However, there is no information given which interpolation method is used for the spatial analyses, e.g. figures 3B, 4, 5 and 6. There is a significant difference for different interpolation methods (e.g. kriging, IDW, linear interpolation. . ..).

———————————————————

---

## Author Comment (AC1) · 8 Jun 2016

nhess-2016-55

Response to Anonymous Referee #1 GIS analysis of effects of future Baltic Sea level rise on the island of Gotland, Sweden Ebert, Ekstedt, Jarsjö

We thank anonymous referee #1 for the comments on our paper. We entirely agree with the referee, the raised points are relevant and interesting!

Study area Gotland: As suggested by the referee, we will improve our motivation why we chose Gotland. Our reason to choose Gotland was that it is the largest island of Sweden, with a fully intact and possible self-maintaining infrastructure. Gotland can only be reached by boat and is an own entity, in comparison to Öland that has a main-

land contact by bridge. Nonetheless, as the referee points out, Öland will suffer even worse from sea level rise, giving its topography, and therefore Öland is our planned next target for future studies of this kind.

We thank the referee for pointing out the ground water quality of the EU water framework – we will include additional information in our revised paper version. Also, as the referee points out, we will make clearer that there is already now a problem with saltwater intrusion and water quality. We would like to point out though that Figure 6 A in our paper shows the current risk of saltwater intrusion.

Discussion: According to the referee's suggestions, we will restructure the discussion. We will discuss life quality and bird life together, and try to reduce the number of subtitle. To meet the criticism of the reviewer we will elaborate more on the economic consequences. However we also wish to point out that economic consequences are complex, long-term, and depending on different possible scenarios both of the effects of sea level rise and the municipality's ways to tackle them. We are in contact with an economist to deep-dive into the economic consequences in a future paper. We will also emphasize that the decrease of natural values is not solely economic.

Comments on technical corrections: Line 104. Referee: County should be country. Gotland is the largest island of Sweden. Yes, correctly it is the largest island of Sweden. However we will not change county to country – Gotland is a county (swed. län) and not a country. County is the correct term.

Line 365 Thank you, yes, entirely correct. We will reword to avoid stating "never before".

Figure 1. Thank you, we will enlarge the location map in the inset, and we will improve the simbols for cities/towns and electricity lines.

Figure 3. Thank you, we will redo the colours for wells at these elvations to better distinguish them.

---

## Author Comment (AC2) · 8 Jun 2016

nhess-2016-55

Response to Anonymous Referee #2

Please note that this is a response to RC2 and also the continuation of RC2 that was attached as a separate file by the reviewer.

We thank anonymous referee #2 for the comments on our paper. In response, we made major revisions that we felt improved the manuscript, including the addition of new analyses showing that our results are robust for reasonable alternative assumptions regarding which weight to assign to different risk factors. Below follows a point-to-point answer to the questions raised.

[Figure]

Rev: The idea is good and the paper points out that the values will be heavily affected due to that much of values are located close to the coastlines. It's interesting with a multi-aspect paper but much is focused on increased salinization of groundwater. Very little data is presented, almost no calculations (. . .). It should have been nice to have a more quantitative estimation of the values or qualitative, pointing out where on Gotland the values are threatened at most. How are different values distributed and connected to the population density?

Response: We regret that the reviewer was under the impression that we present very little data. In response, we now briefly present upfront (already in the end of the introduction) the extensive sets of data that underpin our calculations and interpretations, including (1) well data from 7534 wells including location coordinates, elevation, well depth, groundwater level, and salinity from the Swedish Geological Survey (2) LiDAR elevation data with two-meter resolution over the 3140 km2 study area from the Swedish National Land Survey, (3) meteorological data from the Swedish Meteorological and Hydrological Institute, and (4) other GIS-based data from the County administrative board of Gotland such various infrastructure roads, buildings, wind power stations, power lines, camping sites etc), land cover and land use data (farming land, wetlands, forests, conservation areas, shore meadows, bird nesting areas, endangered plant species habitats) and cultural values (churches, historical wind mills, ancient monuments, etc), see also summary in Table 2 of the manuscript.

In Table 2, we quantify the percentage of land area, infrastructure features, cultural and environmental values that will be directly affected – inundated – by a 2 m sea level rise.

Regarding the suggestion that we should point out where on Gotland the values are threatened at most, we revised the text to clarify that this for instance is seen in Figure 6 (risk of saltwater intrusion by sea level rise), which shows how the different risk values are distributed across Gotland. In addition, Figure 2 shows detailed examples on where nature protection areas, bird protection areas and wells are expected to be inundated due to sea level rise, and Figure 5 illustrates where on Gotland the risk is elevated due

to which risk factor.

Regarding the population density, we do not have access to data on how it varies over Gotland, since the open Swedish data regards average values municipality by municipality, where Gotland is one of the municipalities. (Administratively, Gotland is a special case in Sweden being both a municipality and a county). Nevertheless, the here used approach involving instead well locations and associated spatially distributed well densities enables direct analysis of the extent to which the population in different regions may be affected by salt water intrusion, which is one of our main aims. From a practical viewpoint, well density reflects population density to some extent, since wells are established in contact to human settlements, e.g. location of the larger cities on Gotland (Figure 1) coincide with areas of high well density (Figure 3). As mentioned above, the here used well density data provides sufficient input for the present analysis results.

Rev:...it is important to consider not only land which is inundated (as in the paper) but also land where drainage will be failed leading to less crop production. It is also inadequate just to look at median seawater level since the impact relates to the maximum height of wave actions.

Response: Thank you for this remark. We agree that many factors, including inland inundation due to lack of drainage, would be important to consider in order to understand impacts of future environmental change. However, we here aim at clarifying the relative contribution – and importance – of impacts related to sea level rise. Methodologically, we hence need focus on those effects; otherwise they could not be quantified, and their importance relative to other potential effects could not be distinguished. In the revised manuscript, we now further motivate and clarify the need of taking this approach in order to address the main aims of the study. Our paper is one of many needed to understand the impacts of changing climate; we here focus on sea level rise. Regarding the issue of wave actions, such effects could indeed contribute to the here quantified effects being on the low-end side of the actual outcome, which we now explicitly ac-
knowledge. Nevertheless, our main focus is relative change (future situation relative to today's situation), and we expect that the approach gives robust results in terms of where the risk is higher than elsewhere – although absolute numbers are subject to uncertainty. For instance, wave actions would generally be most inland-effective at the flattest coastal zones, which also would be most affected by the average sea water level rise.

Rev: The novel of the paper is the use of detailed LIDAR data for the GIS evaluation. My concern is regarding the use of information for the RV method. It is not structurized, which it should be according to the principles behind the RV-method, instead uses factors and weights without critical discussion, handling those as if they represent a truth value of salinization risk (se more detailed discussion below). I suggest a major revision of this part and/or present data that support the use and choice of the factors.

Response: We considerably revised the manuscript in the light of these overall comments, which we found to be important. In particular, we now critically discuss the choice of factors, and we additionally test the robustness of our results, by performing and evaluating a new set of analyses based on different (yet still realistic) assumptions regarding risk values. See further our detailed response below.

Rev: L 53-57 The statement that two dm change of the Baltic Sea-level leads to several meters loss of freshwater aquifer is wrong. Ghyben-Herzberg eq cannot be directly applied to the brackish waters in Baltic Sea due to the small density difference where other aspects such as temperature and circulation currents will be more important and lead to a very broad mixed zone. And it is not at all applicable to heterogeneous aquifers (crystalline rocks and sedimentary rocks) prevailing around the Baltic Sea. No reference for the statement is presented.

Response: This is a good point. We removed this hypothetical and simplified example and added instead a reference to a journal paper (Rasmussen et al., 2013) exemplifying in a more nuanced way how future sea level rise can affect the freshwater lens

of a coastal aquifer next to the Baltic Sea. Reference: Rasmussen, P., Sonnenborg, T. O., Goncear, G., & Hinsby, K. (2013). Assessing impacts of climate change, sea level rise, and drainage canals on saltwater intrusion to coastal aquifer. Hydrology and Earth System Sciences, 17(1), 421-443.

Rev: 160- Experience from field measurements indicate that there is not a groundwater sur- face existing on Gotland but a unique groundwater pressure level at each well depend- ing on it's specific depth, geological structures (fractures, stratigraphy, karst) etc. The level varies with season, sometimes up to 20 m. The well data gives probably informa- tion just from the time when the specific well was constructed (I can't find it mentioned in the text), since I assume it is based on the well archieve at SGU. Can we then make a groundwater surface and rely on what it represents?

Response: We thank the reviewer for this statement, that applies to figure 4 A. Correctly there cannot be a connected "groundwater surface". We do regard the information in the figure as highly relevant nonetheless, as the vulnerability of the lowland coastal zones and S Gotland is further stressed. Correctly the well archive shows information from when each well was constructed, which can be seen as snap-shot measurements at different points of time. On the other hand, the depth to groundwater of all our wells does show a rather coherent pattern. To emphasize the uncertainties in this interpolation we will reword the legend and figure text of Figure 4 A to "Estimated depth to groundwater" and Figure 4 B to "Estimated ground water elevation". We will make clear that this is not a continuous groundwater surface.

Rev: The method used refers to the RV-method and the authors explain that the param- eters and the weighting used are similar to what previously been developed for Sweden and Gotland. However, the authors have unfortunately not understood the dynamics of the RV-method. The factor values should be ranked from negative to positive values since it is the extreme conditions that will have a most significant influence on the re- sults. The RV-method also comprises an uncertainty evaluation in the same manner and factors and weighting should be based on a multistatistical evaluation of existing

data from the specific type of terrains that is studied. No such statistical analysis is mentioned. How can we know that the factors selected for Gotland are the most appropriate and that the weighting is statistically correct? The method might have been inspired by the RV-method but cannot be presented as a prolongation of the method.

Response: Thank you – we will reword in the text, to make clear that we use a weighted overlay analysis, which is inspired by the RV method, but does not include its statistical aspects (as the reviewer points out). As also noted by the reviewer in the general comments, we take an appreciated multi-aspect approach to impacts of sea level rise. Hence, since the well salinization issue is one of many aspects covered in the manuscript, we need to keep a reasonable balance regarding level of detail, which explains why we could not fully explore the whole analysis chain associated with the RV-method. Nevertheless, the reviewers question regarding the appropriateness of the considered factors and weights is very relevant. For instance, would the results be similar if much less weight were assigned to some of the factors? In order to check the robustness of our conclusions, we investigated how our results would be influenced if all the factors for simplicity would be given the same weight V=1. This implies very different input in terms of R*V for some key terms, such as the distance to the coastline which had originally V=3. Results showed (see Figure 1 of this response) that the pattern in relative terms remained very similar despite the large differences in assumed R*V. We explain this by the fact that certain regions are subject to high risks due to combined impacts from multiple factors, which means that the results essentially remain the same even in case some of the factors are completely removed from the analysis (or as in the example, by reducing its impact by a factor 3). We also investigated other potential factors including geological ones, and found they would most likely not considerably change the risk amplifications derived from the already considered factors. Finally, we find it interesting that our patterns are much similar to the map of the reviewer's paper in review! Doubtless both papers confirm that the lowland coastlines and especially the southern part of Gotland are present and even worse future risk areas, that need to be considered in Gotland's future planning.

Rev: The papers present five factors included in the risk analysis for saltwater intrusion with reference to a non-published undergraduate report. The choice of factors is therefore unclear.

Response: The undergraduate reports is published and available to the scientific community (we now include its report number in the reference list – it was accidentally omitted in the previous manuscript version). Factors were identified and chosen based on state-of-the art knowledge of their influence on salt water intrusion; however an element of subjectivity can clearly not be avoided in such an approach. We therefore checked the influence of alternative assumptions (and found that absolute numbers are subject to uncertainty although notably our conclusions on relative risk levels are robust). This was explained in detail in the previous paragraph.

Rev: However, a statistical analysis of drilled wells at Gotland made at KTH (Pirnia A, 2012, TRITA-LWR MSc degree project 12:12) shows that two of the variables used in the paper have no statistical significant to chloride wells on Gotland. Two other factors not used in the paper (type of bedrock and landuse) show instead significant correlations to chloride content in the wells. Chloride affected wells are in reality spread all over Gotland since most of the chloride comes from fossil seawater, not from recent seawater intrusion. The method used gives a too high weight to the distance from the shore. A paper submitted to Environmental Earth Sciences shows that the chloride situation on Gotland is much more widespread than is shown in Figure 6 (I enclose a statistical analyses and a GIS map based on the RV-method - not yet scientifically published -cannot be cited)

Response: In the revised manuscript, we will better explain and clarify that our main focus is on determining how much sea level rise can be expected to contribute to well salinization (and other impacts). Hence, in our analysis the distance to the coast, as well as elevation above the sea level, are clearly important factors. Since we primarily emphasize sea level rise effects, factors that show good correlation to well salinization due to other factors, such as intrusion of fossil seawater, would get lower or no weight

in the present study. This may explain part of the differences between our manuscript and the above-mentioned thesis.

Rev: The authors should also refer to the original presentation of the RV method in GIS, which is in Lindberg, Olofsson and Gumbrich (1996) [Lindberg, J., Olofsson, B., Gumbricht,T., (1996): Risk mapping of groundwater salinization using Geographical Information Systems. 14th Salt Water Intrusion Meeting (SWIM), 16-21 June 1996, Malmö, Sweden. SGU Rapporter och Meddelanden no 87, pp 188-197].

Response: Thank you, we will refer to the SWIM paper. Also, we will reword our methods description as to that we do not use a variant of the RV-method, but a weighted overlay analysis which is inspired by the RV-method. Your next point about the freshwater-saltwater transition zone: This is a misunderstanding. We will reword the text to make clear that we mean an inland advancement of the risk of saltwater intrusion in wells and nothing else.

Rev: The figures are generally good and informative. However, there is no information given which interpolation method is used for the spatial analyses, e.g. figures 3B, 4, 5 and 6. There is a significant difference for different interpolation methods (e.g. kriging, IDW, linear interpolation. . .)

Response: In figure 3B, no interpolation method is used, but point density. Point density calculates a magnitude per unit areas from point features that fall within a neighbourhood around each cell. The things that can be changed here is the output cell size, the size of the neighbourhood from which the density is calculated, and the class breaks (in case the density is presented in classes. In figure 3B we chose to present the density as stretched, values 0 (no wells) not displayed). These decisions will change the visual appearance of the result but never change the spots with highest and lowest point density, they will always be the same. The point density function automatically uses nearest neighbour as resampling method.

In figure 4A, we chose IDW to interpolate the estimated depth to groundwater. To

show the overall pattern of many point values, the different interpolation methods show barely any differences. Below figure 2 in this response to illustrate that the result for Kriging are strongly similar to those of IDW.

Figure 5 is simply a presentation of the distribution of values, no interpolation of any values was conducted. Figure 6 is an overlay (an addition of these values in raster calculator), also here, no interpolation.

Figure captions for the two figures below: Figure 1. The overlay analysis of saltwater intrusion risk with unweighted values (figure 6 in our paper shows the same analysis with weighted values for sea level rise). The results of the weighted and unweighted overlay are relatively similar, indicating that the result is robust. Despite the similarities, we argue from a methodological viewpoint to keep our weighted analysis in the main text of the manuscript, since higher values for low elevation above sea level and closeness to the coast reflect known and well-investigated effects sea level rise.

Figure 2. Estimated depth to groundwater interpolated using the KRIGING method.

Legend
**Overlay risk unweighted.tif**
Value

| | |
|---|---|
| ■ | 0 |
| ■ | 1 |
| ■ | 2 |
| ■ | 3 |
| ■ | 4 |
| ■ | 5 |
| ■ | 6 |
| ■ | 7 |
| ■ | 8 |
| ■ | 9 |
| ■ | 10 |

**Fig. 1.**

**Legend**

**Grundvattendjup Kriging_extr 20m dem.tif**
**<VALUE>**

- 0 - 2
- 3 - 3
- 4 - 4
- 5 - 5
- 6 - 7
- 8 - 9
- 10 - 10
- 11 - 13
- 14 - 17
- 18 - 24

**Fig. 2.**

---

## Author Comment (AC3) · 8 Jun 2016

Please find the answer to RC3 (continuation to RC2) in our answer to RC2